# Investigation by DFT Methods of the Damage of Human Serum Albumin Including Amino Acid Derivative Schiff Base Zn(II) Complexes by IR-FEL Irradiation

**DOI:** 10.3390/ijms20112846

**Published:** 2019-06-11

**Authors:** Yuika Onami, Ryousuke Koya, Takayasu Kawasaki, Hiroki Aizawa, Ryo Nakagame, Yoshito Miyagawa, Tomoyuki Haraguchi, Takashiro Akitsu, Koichi Tsukiyama, Mauricio A. Palafox

**Affiliations:** 1Department of Chemistry, Faculty of Science, Tokyo University of Science, 1-3 Kagurazaka, Shinjuku-ku, Tokyo 162-8601, Japan; 1319526@ed.tus.ac.jp (Y.O.); 1315053@ed.tus.ac.jp (R.K.); 1317601@ed.tus.ac.jp (H.A.); b116815@ed.tus.ac.jp (R.N.); 1318623@ed.tus.ac.jp (Y.M.); haraguchi@rs.tus.ac.jp (T.H.); tsuki@rs.kagu.tus.ac.jp (K.T.); 2FEL-TUS, Tokyo University of Science, 2641 Yamazaki, Noda, Chiba 278-8510, Japan; kawasaki@rs.tus.ac.jp; 3Departamento de Química-Física, Facultad de Ciencias Químicas, Universidad Complutense de Madrid, Madrid-28040, Spain; alcolea@quim.ucm.es

**Keywords:** TD-DFT, IR-FEL, human serum albumin, amino acid derivative, Schiff base, Zn(II) complex

## Abstract

An infrared free electron laser (IR-FEL) can decompose aggregated proteins by excitation of vibrational bands. In this study, we prepared hybrid materials of protein (human serum albumin; HSA) including several new Schiff base Zn(II) complexes incorporating amino acid (alanine and valine) or dipeptide (gly-gly) derivative moieties, which were synthesized and characterized with UV-vis, circular dichroism (CD), and IR spectra. Density functional theory (DFT) and time dependent DFT (TD-DFT) calculations were also performed to investigate vibrational modes of the Zn(II) complexes. An IR-FEL was used to irradiate HSA as well as hybrid materials of HSA-Zn(II) complexes at wavelengths corresponding to imine C=N, amide I, and amide II bands. Analysis of secondary structures suggested that including a Zn(II) complex into HSA led to the structural change of HSA, resulting in a more fragile structure than the original HSA. The result was one of the characteristic features of vibrational excitation of IR-FEL in contrast to electronic excitation by UV or visible light.

## 1. Introduction

Against long wavelength UVA light up to about 400 nm, contrary to zinc oxides [1], certain Schiff base Zn(II) complexes incorporating amino acid derivative moiety are expected to have applications for use in sunscreens because they can absorb UVA light [2], thus preventing UVA light from reaching the skin and potentially resulting in many ripple effects. On the other hand, IR light has been reported to damage skin more seriously than UVA light by impacting collagen in the skin’s deep layer [3]. In this way, straightforwardly, one can assume that by absorbing IR light with a sunscreen agent for the IR region, proteins can be protected from serious damages occurring in the skin.

An infrared free electron laser (IR-FEL) can decompose aggregated proteins as well as harmful amyloids by excitation of vibrational bands [4,5,6]. This IR-FEL is a synchrotron radiation-based coherent light with tunable wavelength within the mid-infrared region (5–10 μm) and oscillated at pico-second pulses. Previously, we prepared hybrid materials of protein (human serum albumin; HSA) including a metal complex in order to give additional properties due to metal complexes, for example, photo and redox reactions [7,8]. For photo and redox studies, we prepared the analogous Cu(II) or Zn(II) complexes incorporating amino acids with phenyl moiety [9]. In this study, herein, we synthesized new Schiff base Zn(II) complexes incorporating amino acids with naphthyl moiety or dipeptide derivative with phenyl and naphthyl moieties (Scheme 1; Scheme 2) to increase amide moiety for IR light absorption and compared the degree of damage of Zn(II) complexes, HSA, and hybrid materials of Zn(II) complex-HSA. The mixture of the complex and HSA was irradiated with an IR-FEL [10] and analyzed by infrared microscopic analysis (IRM) by FT-IR. Based on these results, a secondary structure analysis of the protein was performed using Infrared-Secondary Structural Estimation [11], and the structural change of HSA in the time course of IR-FEL irradiation was observed. In order to select IR wavenumbers for irradiation as experimental conditions, experimental IR spectra can suggest absorbing wavenumbers for HSA and Zn(II) complexes and can exhibit actual results for irradiation damage. Additionally, computational methods can support reliable assignments of experimental IR bands for Zn(II) complexes and provide stable conformation (namely some possible optimized structures) of Zn(II) complexes to propose docking features.

## 2. Results and Discussion

### 2.1. Experimental IR Spectra

As stated in the experimental section (as characterization data) later, all five Schiff base Zn(II) complexes had characteristic C=N (imine) bands appear around 1600 cm^−1^, which is a common feature for the analogous Schiff base Zn(II) complexes [9]. Among them, experimental IR spectra for ZnAHN and ZnVHN are depicted in Figure 1; Figure 2, respectively. The bands in these experimental IR spectra indicated that wavelengths of infrared region light were effectively absorbed by the complexes. Detailed assignments of normal vibrations of the Zn(II) complexes are discussed with density functional theory (DFT) calculations later.

On the other hand, the IR spectrum of HSA exhibited many peaks, especially characteristic peaks at about 1650 and 1530 cm^−1^ (not shown). According to advanced research studies dealing with HSA, these peaks can be attributed to amide I and II peaks [12], which are sensitive to structural changes, and also resulting bonding changes of the peptide chains of HSA [13,14] as well as changes by IR-FEL for other proteins [15,16].

### 2.2. DFT Calculations of Zn(II) Complexes

#### 2.2.1. Geometry Optimization in the Isolated State

As for the ZnAHN molecule (Figure 3, left), two possible rotations for the OCH_3_ moiety appear in ZnAHN, according to the value of O22-Zn-O27-C29 dihedral angle −134.3° or −13.7°, Table 1 and Appendix A. Thus, two stable conformers were found, designated for convenience as 1 and 2, respectively (Appendix A). The most stable one, conformer 2 is, however, less stable than conformer 1 when the thermal contribution to the energy is considered (Appendix A). Although the difference of energy between both forms is small, it is expected that in gas phase only conformer 1 appears, and in the solid state both conformers can be present depending on the intermolecular hydrogen bonds with neighbor molecules.

The selected final bond lengths and bond angles optimized in both conformers at the B3LYP/6-31G** level are collected in Appendix A. The labeling of the atoms is plotted in Figure 1 (left) and the discussion below is on both conformers. No crystal structure data are available on ZnAHN.

Conformer 1 appears stabilized by an intermolecular H-bond/contact O22·H28-O27 of 2.480 Å, while conformer 2 is stabilized by the H-bond/contact O22·H32-CH_2_ of 2.544 Å. Conformer 2 appears more stable than conformer 1 because of the slightly shorter O27-Zn bond length, 2.053 Å vs. 2.058 Å in conformer 1, although the intermolecular H-bond/contact O22·H32 is weaker. The main differences between both structures are around the Zn atom, corresponding to the Zn-O and Zn-N bond lengths, O-Zn-O bond angles, and C-C-O-Zn, C-O-Zn-O torsional angles, with a very slight change in their values. The remaining parameters are almost the same between both conformers.

The O22 oxygen appears as the atom with the highest negative charge because it withdraws charge from the Zn at C21 atoms, but it withdraws less negative charge on the Zn atom than it does the O24 oxygen atom. Due to this feature, the Zn-O22 bond length is slightly longer than Zn-O24. The C21 carbon atom has the second most positive charge because O23 and O22 withdraw charge of it. As expected, the highest positive charge corresponds to the Zn atom, 1.27*e* (Appendix A), although this value is noticeably lower than that calculated in the related complex, 1.44*e*.

As for the ZnVHN molecule (Figure 3, right), the replacement of the -C33H_3_ group by the slightly bulkier CH-(CH_3_)_2_ group leads to a slight change in the geometric parameters around the Zn atom. Because the OCH_3_ and the C33H_3_ moieties can rotate, three main stable conformers were optimized in this molecule. Appendix A shows their optimum molecular structure. Thus, the resulting simulated (scaled) IR spectra with assignment of some important peaks for ZnAHN and ZnVHN are depicted in Figure 4 and Figure 5, respectively.

#### 2.2.2. Other Molecular Properties

Calculated wavenumbers were employed to yield thermodynamic properties, which are collected in Appendix A. The theoretical data can be employed to correct experimental thermochemical information at 0 K, as well as for the effect of the zero-point vibrational energy (ZPVE).

To observe the convergence of the energy and thus to analyze the quality of the theoretical results, several basis sets were used. A small increase of energy was observed with the increment of the 6-31G** basis. Therefore, the results obtained in this table with this basis can be considered acceptable. Several thermodynamic parameters, such as enthalpy, heat capacity, free energy, and entropy were calculated for the molecules under study, although only two of them are included. Small differences in the parameters were observed with the increase of the basis set. The entropies calculated for several kinds of compounds and at different ab initio levels have been reported [17] to have mean absolute deviations of less than 5%, as compared to the experimental data. The differences have been attributed to the neglect of residual (orientation) entropy present at 0 K in the crystal.

#### 2.2.3. Scaling the Wavenumbers

To improve the computed wavenumbers, one of the best procedures of scaling was used [18,19,20], i.e., by using two linear scale equations (TLSE). With this procedure of scaling, the error obtained in the scaled wavenumbers is in general lower than 5%, which permits an accurate correlation with the experimental bands and thus their assignments. This procedure of scaling requires the previous calculated wavenumbers of model molecules determined at the same computational level.

This scaling equation procedure, developed by one of the authors, represents a compromise between accuracy and simplicity, and thus it was the only used in the figures for scaling the wavenumbers.

#### 2.2.4. Vibrational Wavenumbers

This study was divided into two regions: 3700–2000 cm^−1^ and 1900–400 cm^−1^, according to the experimental spectra obtained. The scaled IR spectra were simulated using TLSE [20] and they were compared to the experimental FTIR spectra. The calculated, scaled, and experimental values of the ZnAHN molecule in the 3700–400 cm^−1^ range were collected in Appendix A. A resume of the most characteristic bands in the 1800–1200 cm^−1^ range is shown in Table 2. The calculated values were for both conformers, but because of the higher stability at room temperature of conformer 1, its spectrum was analyzed in detail.

Appendix A show the experimental and scaled IR spectra in the 3800–400 cm^−1^ range. Figure 2 shows only the scaled spectrum in the 1800–400 cm^−1^ range, which is the most interesting for the present study. It is noted that the ν(O-H) stretching band appears experimentally as a very broad band and at lower wavenumbers. It can be interpreted that this molecule appears strong H-bonding to O-H group of other molecules, which was not simulated in our calculations. A remarkable discrepancy between theory and experiment appears in the scaled wavenumber of the ν (C=O) stretching mode at 1729 cm^−1^ with very strong IR intensity, with the second one being the most intense in the spectrum, but with surprise, it was not detected experimentally. It can only be interpreted that the experimental results indicate that the molecule does not have this group, or it is strongly H-bonded. This fact, together with other strong discrepancies observed in the comparison of these spectra, suggests a further revision of the experimental structure of the molecule.

A similar scaled spectrum was obtained for the ZnVHN molecule (Figure 3). Their assignments also appear closely to that of ZnAHN and thus they were not included in the present manuscript, although they are available on request. As for ZnGlyGlyH, ZnGlyGlyOH, and ZnGlyGlyPh, conformational effects were quite serious, and simple and reasonable results could not be obtained from DFT calculations.

### 2.3. Damage of HSA (with Zn(II) Complex) by IR-FEL

In the current study, we selected three wavelengths for the IR-FEL irradiation, namely 1652 cm^−1^ corresponding to the C=N (imine) wavenumber of the common Schiff base Zn(II) complex, 1622 cm^−1^ corresponding to the amide I band of HSA, and 1544 cm^−1^ corresponding to the amide II band of HSA. Because of the conformational effects stated in the previous section of DFT results as well as actual compounds, the results for HSA+ZnGlyGlyH, HSA+ZnGlyGlyOH, and HSA+ZnGlyGlyPh are mentioned to a limited extent hereafter, though their whole tendency was similar to that of HSA+ZnAHN and HSA+ZnVHN. All FEL irradiation was carried out with relatively weak power so as not to decompose the Zn(II) complexes and to only decompose HSA.

Figure 6 shows IR spectral changes after FEL irradiation (1652 cm^−1^) plotted by absorbance. Because it is difficult to detect the structural change of has with such plots, we produced and discussed the data after analysis of secondary structural changes (Scheme 3). As we prepared HSA films by casting, the thickness of sample films could not be kept constant. Therefore, qualitative treatment was impossible in this case. However, for the same samples, measurements were carried out before and after irradiation for identical samples, so comparison of intensity of IR spectra for each sample was reliable. We have chosen appropriate types of plots for each IR spectrum in this paper. Overlap of amide bands of different components sometimes results in a poor quality analysis. As depicted in Figure 7, we compared the secondary structural change of the protein itself (only HSA) with that of the hybrid materials with these Zn(II) complexes (HSA+ZnAHN and HSA+ZnVHN) after the FEL irradiation (Figure 7). In the case of irradiation at 1652 cm^−1^ (C=N band), the ratios of β-sheet, β-turn, and other conformations were almost the same in the three samples, although that of α-helix was slightly lower in the two hybrid compounds than that in HSA itself. On the contrary, the protein conformations in the hybrid materials were remarkably changed from the protein itself after irradiation at 1622 cm^−1^: α-helix was increased in both metal complexes, and β-sheet with β-turn and other conformations were substantially decreased in HSA+ZnVHN.

In general, α-helix is frequently found in unaggregated (unbroken) proteins, and β-sheet is frequently found in aggregated (denatured) proteins. According to the irradiation result of 1622 cm^−1^, the Zn(II) complexes are considered to be able to protect proteins. From the results of the HSA+ZnVHN complex, it can be seen that α-helix is small when the protein and the Zn(II) complexes do not coexist, and α-helix increases and β-sheet decreases when the complexes coexist. From this, it is thought that Zn(II) complexes are able to suppress or stabilize the protein structure of HSA. In the case of 1544 cm^−1^, there was almost no change in the two complexes, but it was similar to the case of 1652 cm^−1^ because α-helix decreased and β-sheet increased.

As for dipeptide complexes, only HSA+ZnGlyGlyPh could exhibit a similar tendency mainly because of the conformational effects of ligands (Figure 8). As irradiation time increased, the ratios of α-helix increased and the ratios of β-sheet decreased. It should be noted that the overlap of the C=N band of ZnGlyGlyPh and the amide I band of HSA was avoided by irradiation of 1634 cm^−1^ in this case.

As for HSA+ZnGlyGlyH and HSA+ZnGlyGlyOH, no monotonous increase or monotonous decrease of each secondary structure was observed, which may be attributed to serious conformational changes of these Zn(II) complexes. This fact also supports that the effect of the inclusion of metal complexes is associated with structural (e.g., vibrational) features of protein molecules, which is different from the mechanism of simply absorbing and blocking light that was first expected.

### 2.4. Docking Simulations

As a result of circular dichroism (CD) (partly, not shown) and docking simulation using GOLD (Figure 9), the Zn(II) complexes (ZnAHN, ZnVHN) and HSA were docked well, with scores of 49.9063 and 43.3278, respectively, while the shortest contact distances were 4.938 and 4.211 Å, respectively. Both of the complexes were found to be included inside of HSA molecules, which resulted in slight structural changes of the HSA molecules, even though each complex showed no structural change detected by C=N bond absorption.

As for ZnGlyGlyH, ZnGlyGlyOH, and ZnGlyGlyPh, proposed structures could not be obtained due to multi-conformational problems, as mentioned previously. Thus, we could not carry out docking simulation.

When the HSA-complex was irradiated by IR-FEL at amide I and amide II wavelengths, IR-FEL was irradiated with amide I, amide II wavelength to the HSA-complex complex system, the α-helix significantly changed compared to solely the HSA. Unfortunately, however, DFT calculations about protein molecules as well as hybrid materials of protein-metal complexes may be difficult, and the vibrational analyses of hybrid materials needed to proof this finding would be impossible even in the near future. Contrary to expectations, similar to UV-vis (electronic excitation), the composite of complex/HSA became more unstable to IR (vibrational excitation of IR-FEL) than solely the HSA.

## 3. Materials and Methods

### 3.1. General Procedures

Chemicals of the highest commercial grade available were purchased from Aldrich (St. Louis, MO, USA), Wako (Osaka, Japan), and TCI (Tokyo, Japan) and used as received without further purification.

### 3.2. Preparations

Zn(II) complexes were prepared by common procedures [9] by similarly using the corresponding aldehydes and *L*-amino acids (or gly-gly dipeptide). For example, ZnAHN was synthesized as follows. To a methanol solution (100 mL) of alanine (0.178 g, 2.0 mmol), 2-hydroxy-1-naphthaldehyde (0.3616 g, 2.1 mmol) was added and stirred for 3 hr at 313 K to give rise to a pale yellow solution. Then methanol solution (100 mL) of Zn(OAc)_2_.2H_2_O (0.439 g, 2.0mmol) was added and stirred for 3 hr. The crude compound obtained was filtered, and the precipitate was washed with methanol.

ZnAHN: Yield: 3.7%. UV-vis (0.1 mM MeOH): 240 (π-π*), 310 (n-π*), 380 (CT) nm. Fluorescence: λ_ex_ = 400 nm, λ_em_ = 450 nm. IR (KBr): 1652 (C=N) cm^−1^.

ZnVHN: Yield: 8.4%. UV-vis (0.1 mM MeOH): 240 (π-π*), 380 (CT) nm. Fluorescence: λ_ex_ = 370 nm, λ_em_ = 460 nm. IR (KBr): 1652 (C=N) cm^−1^.

ZnGlyGlyH: Yield: 72%. UV-vis (0.1 mM MeOH): 210 (π-π*), 270 (n-π*), 350 (CT) nm. Fluorescence: λ_ex_ = 370 nm, λ_em_ = 440 nm. IR (KBr): 1634 (C=N) cm^−1^.

ZnGlyGlyOH: Yield: 67%. UV-vis (0.1 mM MeOH): 210 (π-π*), 250 (π-π*), 280 (n-π*) nm. Fluorescence: λ_ex_ = 400 nm, λ_em_ = 450 nm. IR (KBr): 1630 (C=N) cm^−1^.

ZnGlyGlyPh: Yield: 76%. UV-vis (0.1 mM MeOH): 200 (π-π*), 250 (π-π*), 290 (n-π*) nm. Fluorescence: λ_ex_ = 360 nm, λ_em_ = 440 nm. IR (KBr): 1629 (C=N) cm^−1^.

### 3.3. Physical Measurements

Infrared (IR) spectra were recorded by transmission mode using KBr pellets for Zn complex only and by reflection mode as cast films for HSA and HSA+Zn complexes using a stainless plate on a JASCO (Tokyo, Japan) FT-IR 4200 plus spectrophotometer in the range 4000–400 cm^−1^ at 298 K. Electronic (UV-vis) spectra were obtained on a JASCO (Tokyo, Japan) V-570 UV-vis-NIR spectrophotometer in the range 1500–200 nm at 298 K. Fluorescence spectra were measured on a JASCO (Tokyo, Japan) FP-6200 spectrophotometer in the range of 720–220 nm. Circular dichroism (CD) spectra were obtained on a JASCO (Tokyo, Japan) J-820 spectropolarimeter in the range 900–250 nm at 298 K.

### 3.4. Computational Methods

The calculations were carried out by using MP2 ab initio method and by using density functional methods (DFT) [21], including Becke’s three-parameter exchange functional method (B3) [22] in combination with both the correlational functional method of Lee, Yang, and Parr (LYP) [23]. The B3LYP method represents the most cost-effective method [18] and thus it was the only one used in the present manuscript. The B3LYP method was chosen because different studies have shown that the data obtained with this level of theory were in good agreement with those obtained in other more computationally costly methods, such as MP2 calculations, and it predicts vibrational wavenumbers of DNA bases better than the HF and MP2 methods [18,19,24,25]. These methods appear implemented in the GAUSSIAN 09 program package [26]. The UNIX version with standard parameters of this package was running in the alpha computer of the University Complutense of Madrid.

Several basis sets were used, starting from the 6-31G(d,p) to 6-311++G(3df,pd). It was noted that 6-31G(d,p) leads to results that represent a compromise between accuracy and computational cost [20] and thus the results were only presented with this basis set. Low basis sets or poor (approximate) DFT methods for a system can lead to erroneous conclusions, but all our calculations were with the accurate B3LYP method and a large basis set.

The optimum geometry was determined by minimizing the energy with respect to all geometrical parameters without imposing molecular symmetry constraints. Berny optimization under the TIGHT convergence criterion was used. The keyword FREQ was employed for the wavenumber calculations in the harmonic approximation. No imaginary wavenumber was present in the DFT calculated spectra. The natural atomic charges [27] are currently one of the most accurate for correlating properties and for this reason they were the only ones studied in detail. They were determined with the keyword POP=NPA. It should be stressed that the net atomic charges obtained from Mulliken population analysis show extremely strong basis set dependence and they change considerably with the method employed in calculations. Therefore, they were omitted in the present manuscript.

The docking simulation for ZnAHN and ZnVHN was carried out using the GOLD program (CCDC) [28] for ligand (TD-DFT optimized structure)–HSA protein (PDB: 1BM0 crystal structure) docking simulation.

### 3.5. IR-FEL Irradiation

Samples of Zn(II) complexes as KBr pellets and cast films of a HSA membrane and Zn(II)complex-HSA composite membrane were prepared for IR measurements. The formation of composites was confirmed by spectral change of CD spectra but while keeping predominantly secondary structures (not shown). IR-FEL was used at Infrared-Free Electron Laser Research Center of Tokyo University of Science (FEL-TUS) [6,29]. Based on DFT computations, we determined three wavelengths of IR-FEL irradiation: C=N double bond band 6.05 μm (1652 cm^−1^), amide I band 6.16 μm (1622 cm^−1^), and amide II band 6.48 μm (1544 cm^−1^). The intensity of the IR-FEL beam condenses was appropriately tuned in order to avoid breaking chemical bonds in Zn(II) complexes solely for the 30-min irradiation, which was confirmed with IR spectra (not shown) prior to other experiments using HSA. Comparing IR spectra before and after irradiation, changes in each structure (α-helix etc.) were quantified and a protein secondary structure analysis was performed from IR data using analytical software IR-SSE by JASCO (Tokyo, Japan) [30].

## 4. Conclusions

Five new Zn(II) complexes were prepared and characterized with common procedures and docking of the Zn(II) complexes into HSA was confirmed with CD spectra and the GOLD program based on the crystal structure of HSA and the DFT optimized structures of the Zn(II) complexes. Assignment of imine C=N, amide I, and amide II bands (as irradiation wavenumbers at FEL-TUS) could be established to be 1652, 1622, and 1544 cm^−1^, respectively, based on experimental IR spectra or vibrational simulation with TD-DFT calculations. Although at least IR light of 1652 cm^−1^ was well absorbed by both ZnAHN and ZnVHN, the IR spectra of HSA+ZnAHN exhibited little difference with that of HSA for the three wavenumbers, while HSA+ZnVHN was damaged much more than HSA based on the remaining ratio of α-helix in HSA, which suggested that including ZnVHN into HSA led to a structural change of HSA, resulting in a more fragile structure. This result may be a characteristic feature of the vibrational excitation of IR-FEL in contrast to electronic excitation by UV or visible light.

Contrary to the assumption that Zn(II) complexes may act as IR light-absorbing “sunscreens” capable of reducing the damage of protein molecules, the most important observation made during this study is that the inclusion of some metal complex(es) results in it being easy to damage protein molecules by IR-FEL irradiation. Because the biochemical molecules and their hybrid materials as well as some metal complexes exhibiting flexible changes of conformation were not simple, all data could not be obtained under the same conditions. Consequently, possible/impossible or appropriate/ -inappropriate comparisons were described in this paper. Further studies about more simple or rigid Schiff base Zn(II) complexes and other proteins are systematically in progress now.

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
