# Peer review of "Investigation by DFT Methods of the Damage of Human Serum Albumin Including Amino Acid Derivative Schiff Base Zn(II) Complexes by IR-FEL Irradiation"

_ijms, 2019, doi:10.3390/ijms20112846_

Reviewer 1 Report
In this work the authors report a detailed theoretical and experimental investigation on the damage of 2 human serum albumin including amino acid 3 derivative Schiff base Zn(II) complexes. Experiments have been performed by using IR spectroscopy while theoretical computations have been performed at DFT level. In addition docking simulation has been also performed. The subject is interesting, the manuscript well written and the conclusions consistent with the obtained data. The manuscript can be published in the present form
Author Response
Thank you.
The work submitted by the Akitsu group analyses the prospects of application of Schiff base Zn(II) coordination compounds as a novel “sunscreens” that can protect from infrared radiation. As such they inspect how the presence of the Zn-based complexes can protect human serum albumin (HSA) from decomposition/denaturation when exposed to IR radiation originating from free electron laser. As the main methodology of their work, they measure the IR spectrum of different Zn-complexes, HSA, and combination of Zn-complex-HSA combination in the form of KBr pallets. They further apply DFT calculations to describe the IR spectrum of the Zn complexes and finally, they apply docking studies to show the sites at which the Zn-complex binds.
The background idea and choice of Zn-coordination compounds to screen for this application, that is Zn-amino acid complexes, I find reasonable considering low toxicity effects. Further on, the authors provide many details on individual properties such as the IR description by DFT, accounting for most of the absorption modes in great detail. However, I see that the overall work lacks coherence and connectivity between the different parts. Further on the work also lacks a clear conclusion. Besides this I have three main concerns regarding the work from the Akitsu group:
—Section 2.3 discusses the damage on HSA by IR radiation and the role of the Zn-complexes. In essence, this is the core section of the paper. The presented spectra figure 6 only express in arbitrary units. To have a proper evaluation of the spectra and the molecular damage, it is more appropriate to express and to evaluate in terms of the extinction coefficient. The extinction coefficient should be comparable to the remaining spectra of HSA / HSA + Zn-Complex.
—As I understand the preparation method for the IR samples is via KBr pallets. This implies that the compounds have to be smashed in a mortar and then pressed (5-10 bar) into a pallet. This type of IR preparation is excellent for fully inorganic materials, but, from experience with other soft matter/ inorganic-organic hybrids, the smashing can cause “mechanosynthetic” transformations. Can the authors fully exclude that no molecular damage on HSA has been caused during the KBr pallet preparation? In fact, careful examination of the extinction coefficient and parallel evaluation with other IR-based less invasive methods can help in this endeavor.
—If I am not mistaken, from the description by the authors, they do not do any interactions of the Zn-complexes in solution, but they mix the separate compounds in the solid state with KBr and then they make the pallet. If this is correct, then why does one need to have the docking simulation? If there is an interaction, then this interaction is caused by solid material under high pressure (i.e. under mechanochemical condition). Certainly, in the current form, the docking study does account for the experimental scenario and thus does not contribute much to the overall study and understanding.
The authors can certainly do better in their English expression and overall research work presentation delivery (mainly introduction). For instance, it would be advantageous some of the explanations on the different contributions to the peak at ca. 1600 cm-1 of HSA to be introduced in the introduction as well as full coordination formula for each of the 5 Zn-coordination compounds. For better clarity, the authors could have introduced a numerical assignment for their compounds throughout the paper. Finally, I agree with the authors that DFT calculations are prohibitively expensive for calculation of frequency analysis of the full HSA-Zn-complex system, however, they can provide perspective on using approximate DFT methods for this.
Considering the lack of coherency in terms of method choice and conclusions, I do not recommend the current manuscript for publication in IJMS. However, I also think that the authors need to be given a chance for a major revision of their manuscript.
Author Response
Q1 However, I see that the overall work lacks coherence and connectivity between the different parts. Further on the work also lacks a clear conclusion.
A1 Most important observation during this study is inclusion of some metal complex results in being easy to damage protein molecule by IR-FEL irradiation. Because biochemical molecules and their hybrid materials as well as some metal complexes exhibiting flexible changes of conformation were not simple, all data could not be obtained under same conditions. Consequently, possible/impossible or appropriate/not-appropriate comparison were described as mentioned in this paper. This explanation is added in conclusion.
Q2 Section 2.3 ....
A2 As we prepared HSA films by casting, thickness of sample films could not be kept constant. Therefore, qualitative treatment was impossible in this case. However, for the same samples, before and after irradiation measurements were carried out for identical samples, so comparison of intensity of IR spectra for each sample was reliable. We have chosen appropriate types of plots for each IR spectrum in this paper. Moreover, the people work on fullerenes and related systems do it, but in our system I have never seen IR spectra plotted by extinction coefficient. I don’t know the advantages of its use.
Q3 As I understand...
A3 We fixed the description of IR spectroscopy in Experimental Section in order to misunderstanding. KBr pellets were used for samples of Zn complexes. HSA or HSA+Zn complexes were measured as cast films.
Q4 IF I am not....
A4 HSA+Zn complexes samples were not measured as solutions but as cast films. Because it is solid state, docking of HSA+Zn complexes were kept during IR-FEL irradiation and comparable to computational simulation as our other papers about hybrid materials of protein and metal complexes.
Q5 …they can provide perspective on using approximate DFT methods for this….
A5 Low basis set or poor (approximate) DFT method for a system as our, can lead to erroneous conclusions.
That’s all.
Best regards,
Takashiro Akitsu
Department of Chemistry, Faculty of Science, Tokyo University of Science
1-3 Kagurazaka, Shinjuku-ku, Tokyo 162-8601, Japan
Tel. +81-3-5228-8271 (ext. 5775)
Fax. +81-3-5261-4631
E-mail. akitsu2@rs.tus.ac.jp
Round 2
Reviewer 2 Report
Dr. Akitsu and coworkers have provided clarifications to the main questions that I raised in the previous section. It is clear that no invasive methodology is applied during the characterization that can cause significant mechanical damage to the involved HSA species.
However, I still believe that besides the language editing the authors can provide additional improvements to their manuscript. Three important which I find important:
The introduction needs to have clear information on the reported Zn-complexes by the Akitsu group. These are references 7-9 which are just introduced but without clarification. For instance in row 48 the authors write herein, we synthesized new Schiff base Zn(II) complexes incorporating amino acids [9]. the reader may get the impression that these are not new complexes but described before due to the incorrect referencing. When one closely looks at the references will notice that Akitsu and coworkers in this study report derivatives, which then opens the question why these derivatives are more important/considered an improvement over the previously reported in Asian Chem. Lett. 2012, 16, 9‐18.
In the introduction, the authors can add a clear explanation regarding the choice of the methods (theory/experiment) and why it was essential, that is the added value to the study. This can improve the coherence in the chose methods.
In the conclusions, the authors write: "inclusion of some metal complex results in being easy to damage protein molecule by IR‐FEL irradiation." If I am not mistaken, according to the introduction, this is exactly what the authors wanted to prevent by the new molecules? They write: "In this way, straightforwardly, one can assume that by absorbing IR light with a “sunscreen” agent for IR region, proteins can be protected from serious damages occurring in skin." If this is indeed a clear contradiction, it implies that the main hypothesis was disproved or was an incorrect assumption all along. This reduces the rigor of the work, but still needs to be correctly communicated.
Author Response
Please consider the following reply for the second round (minor revison) query by Reviewer 2 and revised manuscript (ijms-513281) uploaded. Correction were highlighted as yellow marks in the revised text.
<Q1> The introduction needs to have clear information on the reported Zn-complexes by the Akitsu group. These are references 7-9 which are just introduced but without clarification. For instance in row 48 the authors write herein, we synthesized new Schiff base Zn(II) complexes incorporating amino acids [9]. the reader may get the impression that these are not new complexes but described before due to the incorrect referencing. When one closely looks at the references will notice that Akitsu and coworkers in this study report derivatives, which then opens the question why these derivatives are more important/considered an improvement over the previously reported in Asian Chem. Lett. 2012, 16, 9‐18..
[A1] According to your comment, description around the row 48 was added to explain and specify the previous related Zn(II) complexes as follows:
“For photo and redox studies, we have prepared the analogous Cu(II) or Zn(II) complexes incorporating amino acids with phenyl moiety [9]. In this study, herein, we synthesized new Schiff base Zn(II) complexes incorporating amino acids with naphthyl moiety or dipeptide derivative with phenyl and naphthyl moieties (Schemes 1 and 2) to increase amide moiety for IR light absorption and compared the degree of damage of Zn(II) complexes, HSA, and hybrid materials of Zn(II) complex-HSA.”
<Q2> In the introduction, the authors can add a clear explanation regarding the choice of the methods (theory/experiment) and why it was essential, that is the added value to the study. This can improve the coherence in the chose methods.
[A2] According to your comment, explanation regarding the choice of the methods was added as follows at the end of introduction: In order to select IR wavenumbers for irradiation as experimental conditions, of course, experimental IR spectra can suggest absorbing wavenumbers for HSA and Zn(II) complexes and can exhibit actual results for irradiation damage. Additionally, computational methods can support reliable assignments of experimental IR bands for Zn(II) complexes and provide stable conformation (namely some possible optimized structures) of Zn(II) complexes to propose docking forms Zn(II) complex in protein molecules.
<Q3> In the conclusions, the authors write: "inclusion of some metal complex results in being easy to damage protein molecule by IR‐FEL irradiation." If I am not mistaken, according to the introduction, this is exactly what the authors wanted to prevent by the new molecules? They write: "In this way, straightforwardly, one can assume that by absorbing IR light with a “sunscreen” agent for IR region, proteins can be protected from serious damages occurring in skin." If this is indeed a clear contradiction, it implies that the main hypothesis was disproved or was an incorrect assumption all along. This reduces the rigor of the work, but still needs to be correctly communicated.
[A3]
We supposed that this “inconsistency” was not mistake nor contradiction, but interesting results beyond initial forecast. Therefore, in the beginning of the second paragraph of conclusion, the following phrase of explanation was added to avoid misunderstanding by readers.
“Contrary to assumption that Zn(II) complexes may act as IR light absorbing “sunscreen” to reduce damage of protein molecules, “
That’s all.
Best regards,
Takashiro Akitsu
Department of Chemistry, Faculty of Science, Tokyo University of Science
1-3 Kagurazaka, Shinjuku-ku, Tokyo 162-8601, Japan
Tel. +81-3-5228-8271 (ext. 5775)
Fax. +81-3-5261-4631
E-mail. <akitsu2@rs.tus.ac.jp>